# New Genera and Species of Trigonidiidae (Orthoptera: Grylloidea) from the Mid-Cretaceous of Myanmar with a Redescription of *Birmaninemobius hirsutus*

**DOI:** 10.3390/insects15060442

**Published:** 2024-06-11

**Authors:** Jun-Jie Gu, Yi Zhou, Wei Yuan

**Affiliations:** College of Agronomy, Sichuan Agricultural University, Chengdu 611130, China; zhy01125@outlook.com (Y.Z.); yuanwei19980808@163.com (W.Y.)

**Keywords:** Kachin, amber, taxonomy, new genus, north Myanmar, morphology

## Abstract

**Simple Summary:**

Orthoptera exhibit rich diversity in Myanmar amber, which is of significant importance for understanding the early evolution and morphology of orthopterans. Here, we report two new genera and three new species of Trigonidiidae (Orthoptera: Grylloidea) from northern Myanmar amber. They cannot be assigned to any subfamily due to their unique combination of characteristics from Trigonidiinae and Nemobiinae. These new findings contribute to expanding the diversity of fossil Trigonidiidae and enhance our understanding of the morphology of the Trigonidiidae.

**Abstract:**

The abundance of insects in Burmese amber illustrates a highly diverse insect community from the mid-Cretaceous period; yet, records of crickets (Grylloidea) are notably scarce. In this study, we describe two new genera with three new species, *Palaeotrigonidium concavoculus* gen. et sp. nov., *Palaeotrigonidium defectivus* sp. nov., and *Tricalcaratus longilineus* gen. et sp. nov., based on three specimens collected in north Myanmar. These new species can be placed within the Trigonidiidae (Orthoptera: Grylloidea) by their triangular head, compound eyes that protrude in dorsal view, and a body entirely covered with robust setae, particularly noticeable in the head and pronotum; however, subfamily assignments are not possible. Another known species, *Birmaninemobius hirsutus*, Xu et al., 2020, from Myanmar amber is redescribed based on a new specimen and a recheck of the holotype.

## 1. Introduction

Trigonidiidae is a large family with wide distribution around the world [1,2]. This family contains two subfamilies, Nemobiinae and Trigonidiinae, including 127 genera and 1039 species, encompassing both extant and extinct species [3,4,5]. Extant Trigonidiidae are characterized by the following characteristics: they have a tiny size and strong setae over the entire body; the frons is broader than the antennal scape; apical and subapical spurs are present; and the metabasitarsomeres are not serrulate except for one apical inner and one apical outer spine [6].

The subfamily Nemobiinae is characterized by a pronotum wider than the head, a protibia with two apical spurs, and a metatibia typically featuring more than three pairs of progressively growing subapical spurs, along with a reduced mirror and needle-shaped ovipositor. Trigonidiinae can be differentiated from Nemobiinae by their pronotum that is smaller than the head, serrated claws on the inner margin, a protibia with a single apical spur, a metatibia featuring three sets of subequal subapical spurs, and a blade-shaped ovipositor [5,6,7].

Up to now, fourteen fossil species of Trigonidiidae have been reported [8,9,10,11,12,13,14]. Among them, the species from the Cretaceous discovered in Myanmar are peculiar, as they exhibit a unique combination of characteristics, cannot be assigned to any subfamily, and probably represent a stem group of Trigonidiidae [15,16,17]. Here, we report two new genera with three species of Trigonidiidae, *Palaeotrigonidium concavoculus* gen. et sp. nov., *Palaeotrigonidium defectivus* sp. nov., and *Tricalcaratus longilineus* gen. et sp. nov, based on three amber specimens from northern Myanmar. We also redescribe *Birmaninemobius hirsutus* Xu et al., 2020 based on a new specimen from northern Myanmar and a re-examination of the holotype.

## 2. Materials and Methods

The specimens were deposited at the Department of Plant Protection of Sichuan Agricultural University (SICAU), Chengdu, China, before 2017 (Jun-Jie, Gu, Curator). In this study, all ambers were collected from the Hukawng Valley, Myitkyina District, Kachin Province, in northern Myanmar [18]. Burmese amber was dated to the earliest Cenomanian (98.79 ± 0.62 Ma) based on the UePb dating of zircons from the volcaniclastic matrix of the ambers [19] or to the late Albian–early Cenomanian based on biostratigraphic and radioisotope data [20,21,22,23].

The amber containing each specimen was ground and polished to a suitable size. Photographs were taken with an SZX16 microscope system and cellSens Dimension 3.2 software (Olympus, Tokyo, Japan). In most instances, incident and transmitted light were used simultaneously. All the images were digitally stacked in photomicrographic composites of approximately 30 individual focal planes obtained using Helicon Focus 6 (https://www.heliconsoft.com/ accessed on 12 May 2022) for a better illustration of the 3D structures.

The morphological terminology generally followed that established by Otte and Alexander with minor modifications by Heads and Heads et al. [24,25,26].

The terminology of the fore wing venation followed that proposed by Béthoux [27], Campos et al. [28], and Josse et al. [29]. Sc, subcosta; R, radius; M, media; MP, posterior media; CuA, anterior cubitus; CuP, posterior cubitus; CuPa, first posterior cubitus; CuPb, second posterior cubitus; PCu, postcubital; PCuA, anterior postcubital; PCuP, posteriorpostcubital; A, anals; AA, anterioranals; AP, posterioranals; d, dividing vein; di, diagonal vein; ch, chord vein; and hv, harp vein.

## 3. Results

Systematic paleontology.

Order Orthoptera Olivier, 1789.

Suborder Ensifera Chopard, 1920.

Infraorder Gryllidea Laicharting, 1781.

Superfamily Grylloidea Laicharting, 1781.

Family Trigonidiidae Saussure, 1874.

Genus *Palaeotrigonidium* gen. nov.

Included species. *Palaeotrigonidium concavoculus* sp. nov. (type species), *Palaeotrigonidium defectivus* sp. nov (Figure 1A,B).

Diagnosis. The genus differs from other Trigonidiidae genera reported in Burmese ambers by the presence of a protibia with two apical spurs (Figure 2C,D); furthermore, *Palaeotrigonidium* gen. nov. differs from *Curvospurus* He, 2020 (two inner and three outer spurs), *Qiongqi* Yuan, Ma & Gu, 2023 (three inner and four outer spurs), and *Tricalcaratus* gen. nov. (four inner and four outer spurs) by the metatibia with six subapical spurs (three inner and three outer spurs) (Figure 2E,F). It also differs from *Curvospurus* He, 2020, *Qiongqi* Yuan, Ma & Gu, 2023, and *Tricalcaratus* gen. nov. (which have six apical spurs) by the metatibia with five apical spurs (Figure 2E,F) and from *Birmaninemobius* Xu et al., 2022, by the metabasitarsus armed with serrulations (Figure 2E,F).

Etymology. The generic name derives from the combination of the Latin word ‘*Palaeo*’, meaning of ancient, and ‘*trigonidium*’, the usual suffix given to Trigonidiine crickets.

*Palaeotrigonidium concavoculus* sp. nov.

Material. Holotype, SICAU(A)-144. Female, Burmese amber (Myanmar, Kachin Province, Hukawng Valley); mid-Cretaceous, latest Albian to earliest Cenomanian; a complete adult insect with prothoracic legs, mesothoracic legs, and metathoracic legs; the whole body is covered by abundant bubbles, the ovipositor is complete.

Diagnosis. Compound eyes developed and droplet-shaped; it differs from *Palaeotrigonidium defectivus* sp. nov. by its winglessness, a protibia with only an inner tympanum (Figure 2C,D), and an abdominal sternite with alternating dark and light spots.

Etymology. The specific epithet derives from a combination of the Latin words ‘*concavus*’, meaning concave, and ‘oculus’, meaning eye, describing the compound eyes slightly concave in shape.

Description. Holotype, SICAU(A)-144. Female.

Head, triangular; vertex, broad, with a series of long setae; frons, flat, wider than the antennal scape, with several sparse setae; compound eyes developed and droplet-shaped, situated near the dorsal surface of the head, obviously protruding from the head; ocelli present, with two lateral ocelli settled behind the scape and one median ocellus in the middle of the scape; antennal scape, shield-like, located between the compound eyes; antennae, filiform, incomplete; maxillary palpus, slender, end joint with a slightly broad terminal, labial palpi relatively short; clypeus covering the mandible; maxillae, slender and longer than the mandible.

Pronotum, wider than long, narrower than the head in the dorsal view, with a series of long setae along the anterior, posterior, and lateral margins, several dark marks on the anterior and posterior margins, the dorsal disc with several sparse setae, and the lateral lobe with abundant hairs.

Prothoracic legs, comparatively short and thin, covered by fine setae, with several dark bands spaced at intervals; procoxa, cylindrical; protibia with an oval tympanum on the inner side and two apical spurs; probasitarsus, parallel and ventrally covered by two rows of short setae; second tarsomere, short; third tarsomere, shorter than the basitarsus, with two slightly curved claws on the apex.

Mesothoracic legs similar to the prothoracic legs; mesotibia with three apical spurs on the apex.

Metathoracic legs, long and thin, covered with hairs and setae; metafemur with several dark marks and a band near the distal margin; metatibia slightly shorter than the metafemur, with several dark bands spaced at intervals, featuring a row of subapical spurs on both inner and outer dorsal margins (three inner and three outer spurs), along with five apical spurs on the apex (two inner three outer spurs); metabasitarsus, serrulate, covered with setae, with two apical spurs on the apex; second tarsomere, short, third tarsomere with two slightly curved claws.

Abdomen, stubby, with subgenital plate slightly concave at the distal end; cerci, incomplete, with fine setae; epiproct, rectangular, ovipositor, needle-like with a pointed terminal.

The measurements were as follows: body length, 4.57 mm (measured from the head to the abdominal apex); head length, 1.07 mm; ovipositor length, 2.12 mm; profemur length, 1.01 mm, protibia length, 0.96 mm; mesofemur length, 0.95 mm, mesotibia length, 0.90 mm; metafemur length, 2.59 mm; and metatibia length, 1.95 mm.

Remarks. Due to the absence of the metatibia in *Palaeotrigonidium defectivus* sp. nov., we were unable to compare the number and condition of the subapical and apical spurs in these two species. However, based on the droplet-shaped compound eyes and the presence of only the inner tympanum in the protibia (Figure 2A–D), we excluded the possibility that this specimen is a female individual of *P. defectivus* sp. nov.

*Palaeotrigonidium defectivus* sp. nov.

Material. Holotype, SICAU(A)-008. Male, Burmese amber (Myanmar, Kachin Province, Hukawng Valley); mid-Cretaceous, latest Albian to earliest Cenomanian; a nearly complete adult insect with antennae, protibiae, mesotibiae, and metafemur; tegmina, well preserved.

Etymology. The specific epithet is from the Latin word ‘*defectivus*’, meaning defective, describing the incomplete preservation state of the legs (Figure 3A,B and Figure 4C,D).

Diagnosis. Protibia with tympana on both sides (Figure 4C–F); tegmina with two oblique veins, CuPb vein fused with the proximal oblique vein.

Description. Holotype, SICAU(A)-008. Male.

Head, triangular; vertex, broad, adorned with a series of long setae; frons, flat, wider than the antennal scape, with several sparse setae; compound eyes developed and round-shaped, situated near the dorsal surface of the head, obviously protruding from the head; ocellus, invisible; antennal scape, cylindrical, located between the compound eyes, antennae, filiform, incomplete; maxillary palpus, slender, end joint, slightly broad with a transverse terminal, labial palpi, relatively short.

Pronotum, wider than long, narrower than the head in dorsal view, with dark marks on the posterior and lateral margins; anterior, posterior, and lateral margins with a series of long setae; dorsal disc with several sparse setae, lateral lobe with abundant hairs.

Profemur and protibia with fine setae; protibia with oval tympana on both sides and two apical spurs; tarsi not preserved.

Mesothoracic legs similar to prothoracic legs; mesotibia with three apical spurs on the apex; tarsi not preserved.

Metathoracic legs incomplete, with only the metafemur preserved, covered with some dark marks, and a wide ventral gutter, largely extended laterally on the inner side.

Tegmina, present and developed; tegmina, shorter than the abdomen, more or less truncate posteriorly, basal field, longer than pronotum; Sc vein with five branches in the lateral field; media vein fused with the CuA vein; CuPb vein fused with the proximal oblique vein; two oblique veins present; proximal vein, shortest, apical vein, longest; all oblique veins located between the PCuA vein and the CuPa vein; CuPa vein forked into CuPaα vein and CuPaβ vein at the outer corner of the mirror, and CuPaα vein forked into CuPaα1 vein and CuPaα2 vein; diagonal vein, short and almost straight, linked with PCuA vein and CuPaβ vein; PCuP vein and AA vein fused with PCuA vein at the apex of the basal field, slightly before the intersection of diagonal vein, PCu veins, and AA vein; AP vein, sinuous, very close to AA vein, ended at the inner margin of the apex of the basal field. Three chord veins present: inner one (PCuA vein), longest and mostly arc-like, linked to the mirror by an almost straight transverse vein; middle one (extension of PCuP vein), bisinuate; and outer one (extension of AA vein), shortest and almost straight; mirror, large and elongated, about half the length of the tegmina, armed with an acute anterior corner and a somewhat rounded posterior corner. Dividing vein curved, dividing the mirror into two portions, with the anterior part larger than the posterior one; apical field, very short, with a band-like cell along the posterior margin of the mirror.

Abdomen, mostly obscured by bubbles, cerci incomplete, adorned with fine setae.

The measurements were as follows: body length, 4.19 mm (measured from the head to the abdominal apex); head length, 0.61 mm long; tegmina length, 2.19 mm; profemur length, 1.22 mm; protibia length, 0.86 mm; mesofemur length, 1.11 mm; mesotibia length, 0.93 mm; and metafemur length, 2.97 mm.

Genus *Tricalcaratus* gen. nov.

Type species. *Tricalcaratus longilineus* sp. nov. (Figure 5A,B)

Etymology. The generic name derives from a combination of the Latin words ‘*tres*’, meaning three, and ‘*calcar*’, meaning spur, describing the protibia with three spurs.

Diagnosis. Tegmina with four oblique veins, CuPb vein fused with CuPa vein at the corner of the mirror; hindwings, long, extended over the tegmina. The genus differs from other Trigonidiidae genera reported in Burmese ambers by the presence of a protibia with three apical spurs and a metatibia with eight subapical spurs (Figure 6A,B,E,F); furthermore, this genus differs from *Birmaninemobius* Xu et al., 2022, and *Palaeotrigonidium* gen. nov. (all with five apical spurs) by the presence of a metatibia with six apical spurs (Figure 6E,F). It differs from *Birmaninemobius* Xu et al., 2022, by the presence of a metabasitarsus armed with serrulations (Figure 6E,F).

*Tricalcaratus longilineus* sp. nov.

Material. Holotype, SICAU(A)-146. Male, Burmese amber (Myanmar, Kachin Province, Hukawng Valley); mid-Cretaceous, latest Albian to earliest Cenomanian; a complete adult insect with prothoracic legs, mesothoracic legs, metathoracic legs, tegmina, and hindwings well preserved.

Etymology. The specific epithet derives from the Latin word ‘*longilineus*’, meaning “having a long, slender form”, describing its long body.

Diagnosis. As for the genus.

Description. Holotype, SICAU(A)-146. Male, body form, slender.

Head, triangular, with a broad vertex adorned with a series of setae; frons, narrow, slightly protruding, featuring several sparse setae; compound eyes developed and ellipse-shaped, situated near the dorsal surface of the head and obviously protruding from the head; ocellus, invisible; antennal scape shield-like, located between the compound eyes, antennae, filiform, incomplete; maxillary palpus, obviously slender, with end joint with a slightly broad terminal, labial palpi relatively short; clypeus covering the mandible; both maxillae and mandible, long and slender (Figure 7A,B).

Pronotum, transversal, narrower than the head in dorsal view, with a series of long setae along the anterior, posterior, and lateral margins, disc with several sparse setae and several dark marks, lateral lobe adorned with abundant hairs.

Prothoracic legs comparatively short and thin, profemur and protibia with fine setae; procoxa, cylindrical; protibia with three apical spurs, with an inner tympanum (Figure 7E,F); probasitarsus, parallelly and ventrally covered by two rows of short setae; second tarsomere, short; third tarsomere, shorter than the basitarsus, with two slightly curved claws.

Mesothoracic legs, similar to the prothoracic legs; mesotibia with three apical spurs on the apex.

Metathoracic legs, long and thin, covered with hairs and setae; metatibia, slightly shorter than the metafemur, featuring a row of subapical spurs on both inner and outer dorsal margins (four inner and four outer spurs), along with six apical spurs on the apex (three inner three outer spurs); metabasitarsus, serrulate, covered with setae, with two apical spurs on the apex; second tarsomere, short, third tarsomere with two slightly curved claws.

Both tegmina and hindwings present and developed; tegmina elongated, with the basal field longer than the pronotum; Sc vein with six branches in the lateral field; media vein fused with CuA vein; CuPb vein fused with CuPa vein at the corner of the mirror; four oblique veins present, proximal vein shortest, apical vein longest; three oblique veins positioned between PCuA vein and CuPa vein, with CuPb vein crossing by, and apical vein located between PCuA vein and CuPb vein; CuPa vein forked into CuPaα vein and CuPaβ vein at the outer corner of the mirror, then CuPaα vein further forked into CuPaα1 vein and CuPaα2 vein; diagonal vein, short and almost straight, linked to PCuA vein and CuPaβ; PCuP vein and AA vein fused with PCuA vein at the apex of the basal field, slightly before the intersection of diagonal vein, PCu veins, AA vein. Three chord veins present: inner one (PCuA vein), longest, mostly arc-like, connected with the mirror by an almost straight transverse vein; middle one (extension of PcuP vein), bisinuate; and outer one (extension of AA vein), shortest and almost straight; mirror, large and elongated, about one-third the length of the tegmina, armed with an acute anterior corner and a somewhat rounded posterior corner; dividing vein, nearly straight, dividing the mirror into two portions, with the posterior part larger than the anterior one; mirror with a band-like cell along the posterior margin, apical field covered by bubbles, lost some parts due to preservation; hindwings, long, surpassing the apex of the abdomen, covered by the tegmina, largely invisible.

Abdomen, elongated, subgenital plate slightly concave at the distal end, cerci rather long, with fine setae.

The measurements were as follows: body length, 5.79 mm (measured from the head to the abdominal apex); head length, 1.72 mm long; tegmina length, 4.01 mm; hindwings length, 7.41 mm; profemur length, 1.03 mm; protibia length, 1.36 mm; mesofemur length, 1.08 mm; mesotibia length, 1.62 mm; metafemur length, 3.17 mm; and metatibia length, 3.42 mm.

Genus. *Birmaninemobius* Xu et al., 2020.

Type species. *Birmaninemobius hirsutus* Xu et al., 2020. (Figure 8A,B)

Amended diagnosis. Ectoparamere flake-shaped, slightly curved, with two pointed tips at the end in ventral view, guiding rod, long with slightly widened end. The genus differs from other Trigonidiidae genera reported in Burmese ambers by the presence of non-serrulate metabasitarsi (Figure 9E,F); it differs from *Curvospurus* He, 2020, *Qiongqi* Yuan, Ma & Gu, 2023, and *Tricalcaratus* gen. nov. (all with six apical spurs) by the presence of a metatibia with five apical spurs (Figure 9G,H), from *Curvospurus* He, 2020 (four apical spurs), *Palaeotrigonidium* gen. nov. (two apical spurs), and *Tricalcaratus* gen. nov. (three apical spurs) by the presence of a protibia with only one apical spur (Figure 9A,B), and from *Curvospurus* He, 2020 (two inner and three outer spurs), *Qiongqi* Yuan, Ma & Gu, 2023 (three inner and four outer spurs), and *Tricalcaratus* gen. nov. (four inner and four outer spurs) by the presence of a metatibia with six subapical spurs (three inner and three outer spurs) (Figure 9E,F).

*Birmaninemobius hirsutus* Xu et al., 2020.

Material. SICAU(A)-163. Male, Burmese amber (Myanmar, Kachin Province, Hukawng Valley); mid-Cretaceous, latest Albian to earliest Cenomanian; a complete adult insect with prothoracic legs, mesothoracic legs, metathoracic legs, and tegmina well preserved.

Redescription. Head, triangular; vertex, broad, adorned with a series of long setae; frons, flat, wider than the antennal scape, with several sparse setae; compound eyes developed and oval-shaped, situated near the dorsal surface of the head, protruding from the head; ocellus, invisible; antennal scape, cylindrical, located between the compound eyes, antennae, filiform, incomplete.

Pronotum, transversal, narrower than the head in dorsal view, with a series of long setae along the anterior, posterior, and lateral margins, dorsal disc, light in color, with several sparse setae, lateral lobe with abundant hairs and some dark marks.

Prothoracic legs, comparatively short and thin, profemur and protibia with fine setae and several dark bands spaced at intervals; procoxa, cylindrical; protibia with one apical spur, inner tympanum (Figure 10A,B), present; probasitarsus, parallel and ventrally covered by two rows of short setae; second tarsomere, short; third tarsomere, shorter than the basitarsus, with two slightly curved claws.

Mesothoracic legs, similar to prothoracic legs; mesotibia with two apical spurs on the terminal.

Metathoracic legs, long and thin, with several dark bands spaced at intervals, covered with hairs and setae; metatibia, slightly shorter than the metafemur, featuring six subapical spurs (three inner three outer), along with five apical spurs on the apex; metabasitarsus, non-serrulate, covered with setae, with two apical spurs on the apex; second tarsomere, short, and third tarsomere with two slightly curved claws.

Tegmina present and developed; tegmina, shorter than the abdomen, more or less truncate posteriorly, basal field, longer than the pronotum; media vein fused with CuA vein; CuPb vein fused with CuPa vein before the harp; two oblique veins present, almost of the same length; all oblique veins located between PCuA vein and CuPa vein; CuPa vein forked into CuPaα vein and CuPaβ vein at the outer corner of the mirror, CuPaα vein further forked into CuPaα1 vein and CuPaα2 vein; diagonal vein, short, almost straight, linked to PCuA vein and CuPaβ; PCuP vein and AA vein fused with PCuA vein at the apex of the basal field, slightly before the intersection of diagonal vein, PCu veins, and AA vein, AP vein sinuous, fused with AA vein. Three chord veins present: inner one (PCuA), longest and mostly arc-like, connected with the mirror by an almost straight transverse vein, middle one (extension of PcuP vein), bisinuate, and outer one (extension of AA vein), shortest and almost straight. Mirror, large and elongated, armed with an acute anterior corner and a somewhat rounded posterior corner. Dividing vein, curved, dividing the mirror into three portions, with the longest one intersecting with CuPaβ vein; apical field, very short, with a band-like cell along the posterior margin of the mirror.

Abdomen, stubby, epiphallus, broad, lateral lobes, bluntly rounded at the distal end; ectoparamere, flake-shaped, slightly curved downwards, with two pointed tips in ventral view; guiding rod, long with a slightly widened apex; cerci rather elongated, with fine setae.

The measurements were as follows: body length, 4.27 mm (measured from the head to the abdominal apex); head length, 0.56 mm; tegmina length, 1.83 mm; profemur length, 1.09 mm; protibia length, 1.09 mm; mesofemur length, 1.03 mm; mesotibia length, 1.03 mm; metafemur length, 2.36 mm; and metatibia length, 1.80 mm.

Remarks. *B. hirsutus* was initially assigned to Nemobiinae by the presence of a compressed second metatarsomere without adhesive pads (neither widened nor flat) and serrated claws and of a mirror similar to a common cell of the forewing’s apical field [16]. However, based on the triangular head, a longer than broad scape, and a protibia bearing just one long apical spur, Desutter-Grandcolas, et al. revised this classification and suggested that this species should be described as belonging to a stem group of Trigonidiidae [6]. We examined the holotype and the new specimen and found that some parts of the original description were inaccurate, such as those regarding the shape of the mirror and the presence of a median ocellus and the caption of the terminal abdominal (Figure 10C–F and Figure 11A–F). The specimen SICAU(A)-163 exhibits compound eyes that appeared more rounded and less pronounced compared to those of NIGP172331. Additionally, the ectoparamere of SICAU(A)-163 appeared more hook-like, while that of NIGP172331 appeared more flake-like (Figure 11E,F). It is hard to accurately assess the morphological differences of this skeletal structure in ambers; meanwhile, some membranous structures of the genitalia are often difficult to preserve. Furthermore, these differences may also be attributed to variations in photography angles and preservation conditions. However, as no distinct differences were observed in the apical spurs of the protibia, the apical and subapical spurs of the metatibia, the wing venation, and other features, we decided to temporarily consider it as a new material of *B. hirsutus*.

## 4. Discussion

These newly found species can be attributed to the family Trigonidiidae based on the following characteristics: compound eyes protruding in dorsal view; and entire body covered with strong setae, especially on the head and pronotum. *Palaeotrigonidium concavoculus* sp. nov. shares some characteristics with species of the subfamily Trigonidiinae, such as a triangular head, a pronotum narrower than the head in dorsal view, and a metatibia with six apical spurs (three inner and three outer); it presents a lateral ocellus, a protibia with two apical spurs, a simple and non-serrated claw, normal second tarsomeres (neither widened nor flat), and a needle-like ovipositor. These characteristics contrast with those of the subfamily Nemobiinae. *Palaeotrigonidium concavoculus* sp. nov. can be distinguished from species of these two subfamilies by its unusual shape of compound eyes and the serrulate metabasitarsomere. *Palaeotrigonidium defectivus* sp. nov. shares certain characteristics with species of the subfamily Trigonidiinae, such as a triangular head, wider than the pronotum in dorsal view, and the absence of a lateral ocellus; the new species is similar to species of the subfamily Nemobiinae for the presence of two apical spurs on the terminal of the protibia. *Tricalcaratus longilineus* sp. nov. is similar to species of the subfamily Trigonidiinae, with a triangular head, wider than the pronotum in dorsal view. The new species shares characteristics with species of the subfamily Nemobiinae, such as non-serrated simple claws, non-specialized second tarsomeres, and more than three pairs of subapical spurs. *Tricalcaratus longilineus* sp. nov. differs from species of these two subfamilies for its unusual head (with a relatively narrow frons and slender mandible and maxillae), a protibia with three apical spurs, and serrulate metabasitarsomere. Thus, these new species cannot be assigned to a Trigonidiidae subfamily.

Currently, ambers with Trigonidiidae have been reported from various geological periods, including the mid-Cretaceous, the Oligocene, the middle-late Eocene, the Eocene, and the Miocene. Apart from the Trigonidiidae insects found in mid-Cretaceous Burmese amber, most fossil species can be classified into the subfamilies Trigonidiinae and Nemobiinae or into a specific extant genus [8,9,10,11,12,13,14]. The specimens reported from Burmese ambers, *Birmaninemobius hirsutus*, *Curvospurus huzhengkun* He, 2020, and *Qiongqi crinalis* Yuan, Ma & Gu, 2023, also show features of both Nemobiinae and Trigonidiinae such as a pronotum smaller than the head, non-serrated claws, and the second tarsomerses that are neither widened nor flat. At the same time, Trigonidiidae from Burmese ambers also showed unique features that could not be observed in extant species. In extant Trigonidiidae species, the number of apical spurs on the protibia is relatively stable. However, in Burmese amber species, such as *Curvospurus huzhengkun* He, 2020, with four apical spurs on the protibia, and *Tricalcaratus longilineus* sp. nov., with three apical spurs, the spur number exceeds those observed in extant Trigonidiidae insects. Additionally, *Curvospurus huzhengkun* He, 2020, *Qiongqi crinalis* Yuan, Ma & Gu, 2023, *Palaeotrigonidium concavoculus* sp. nov., and *Tricalcaratus longilineus* sp. nov. have a serrulate metabasitarsus, which is distinct from that of extant Trigonidiidae. Over an extended period, the classification of Orthoptera has predominantly relied on morphological characteristics, with certain features being defined rather subjectively and lacking support from a phylogenetic framework, and the descriptions of many new species often exhibit mutual contradictions [30]. Currently, although the molecular data-based reconstruction of the phylogeny of Orthoptera has gained widespread acceptance [1,28,31,32], stable morphological synapomorphies are still lacking, preventing the definition of distinct clades among various groups. Desutter-Grandcolas et al. re-analyzed ambers with Trigonidiidae species and redefined the morphological diagnoses of Trigonidiidae [6]. However, with the publication of more amber species, some features appear to be less applicable to certain fossil groups. Therefore, the continuous discovery of fossil species is crucial for understanding the systematics and morphological evolution of the Trigonidiidae, all the Grylloidea, and even the Orthoptera.

## 5. Conclusions

Based on the morphological analysis described above, two new genera and three new species, i.e., *Palaeotrigonidium concavoculus* gen. et sp. nov., *Palaeotrigonidium defectivus* sp. nov., and *Tricalcaratus longilineus* gen. et sp. nov., were reported here. We redescribed the species *Birmaninemobius hirsutus* Xu et al., 2020, based on a new specimen from northern Myanmar and the examination of the holotype. These new findings increase the diversity of fossil Trigonidiidae and provide new knowledge on their morphology.

## Figures and Tables

**Figure 1 insects-15-00442-f001:**
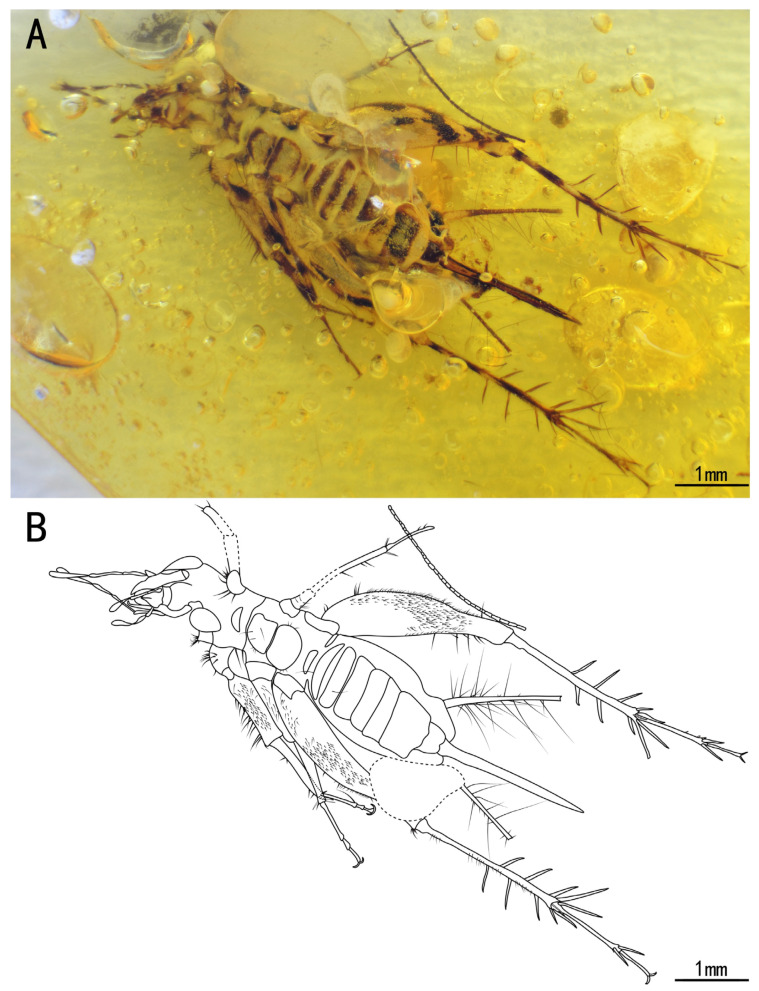
Habitus of *Palaeotrigonidium concavoculus* sp. nov.: (**A**) photograph of ventral view and (**B**) line drawing of ventral view. Dotted line indicates obscured parts.

**Figure 2 insects-15-00442-f002:**
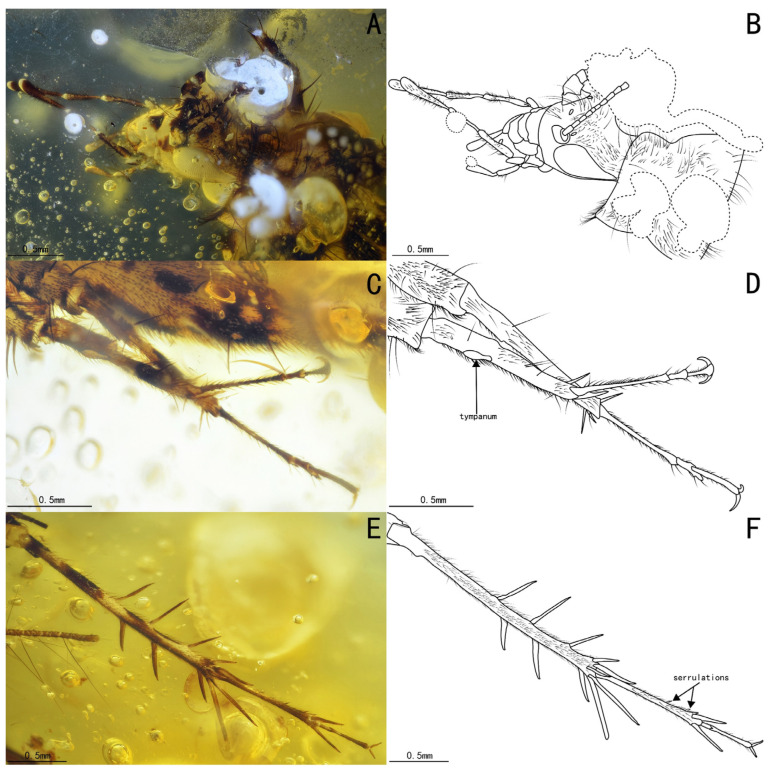
Details of *Palaeotrigonidium concavoculus* sp. nov.: (**A**) photograph of the head, (**B**) line drawing of the head, (**C**) photograph of the prothoracic leg and mesothoracic leg, (**D**) line drawing of the prothoracic leg and mesothoracic leg, (**E**) photograph of the metathoracic leg, and (**F**) line drawing of the metathoracic leg. Dotted line indicates obscured parts.

**Figure 3 insects-15-00442-f003:**
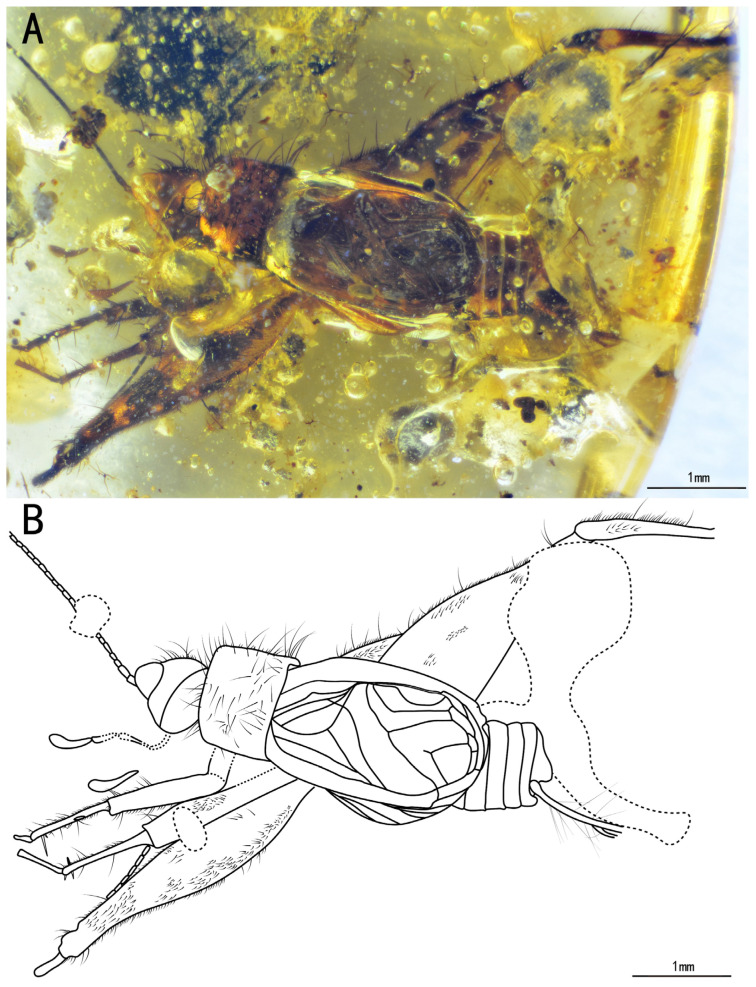
Habitus of *Palaeotrigonidium defectivus* sp. nov.: (**A**) photograph of dorsal view and (**B**) line drawing of dorsal view. Dotted line indicates obscured parts.

**Figure 4 insects-15-00442-f004:**
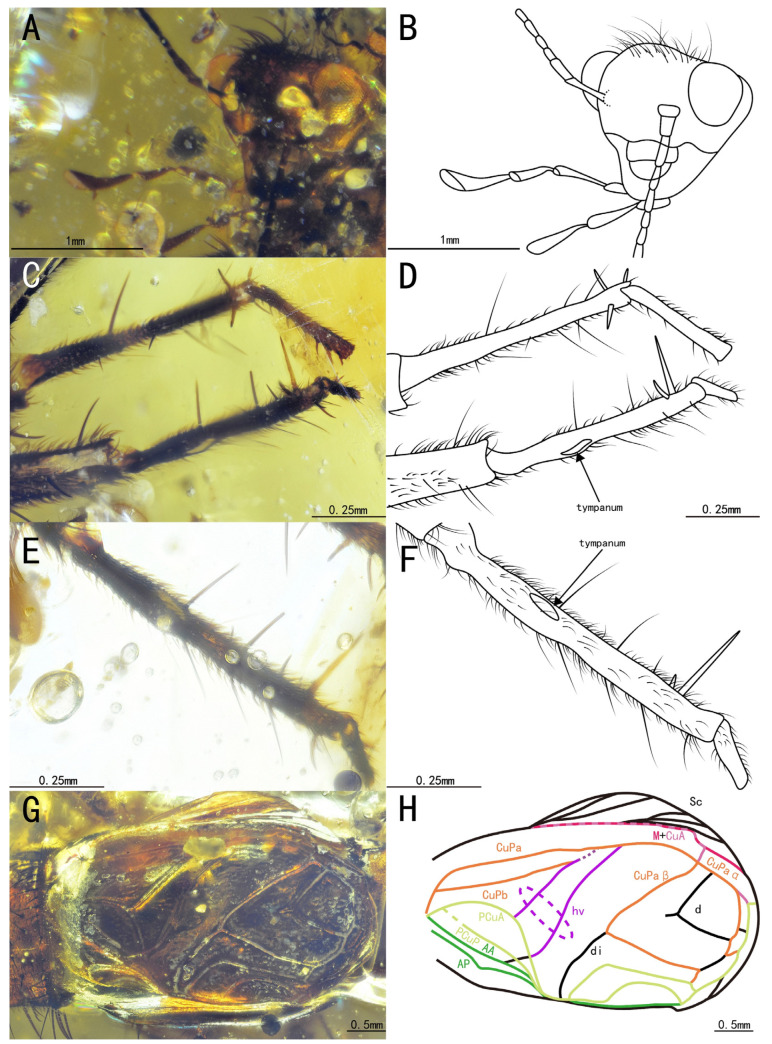
Details of *Palaeotrigonidium defectivus* sp. nov.: (**A**) photograph of the head, (**B**) line drawing of the head, (**C**) photograph of the prothoracic leg and mesothoracic leg, (**D**) line drawing of the prothoracic leg and mesothoracic leg, (**E**) photograph of the prothoracic leg to show the outer tympanum, (**F**) line drawing of the prothoracic leg to show the outer tympanum, (**G**) photograph of the tegmina and (**H**) line drawing of the tegmina.

**Figure 5 insects-15-00442-f005:**
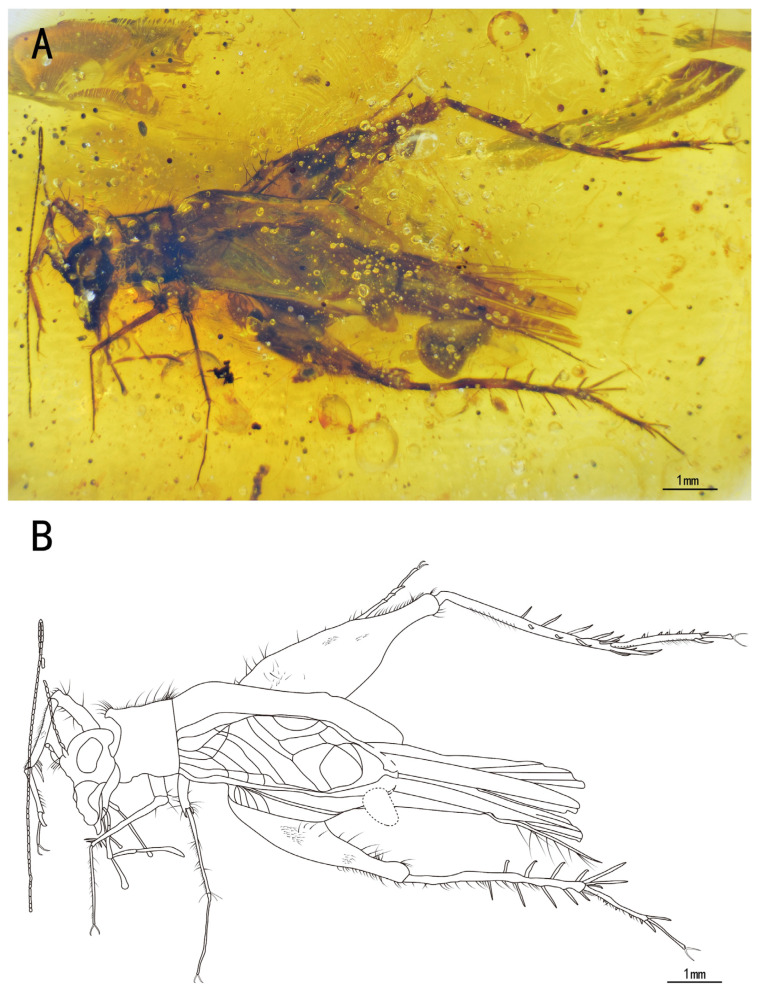
Habitus of *Tricalcaratus longilineus* sp. nov.: (**A**) photograph of dorsal view and (**B**) line drawing of dorsal view. Dotted line indicates obscured parts.

**Figure 6 insects-15-00442-f006:**
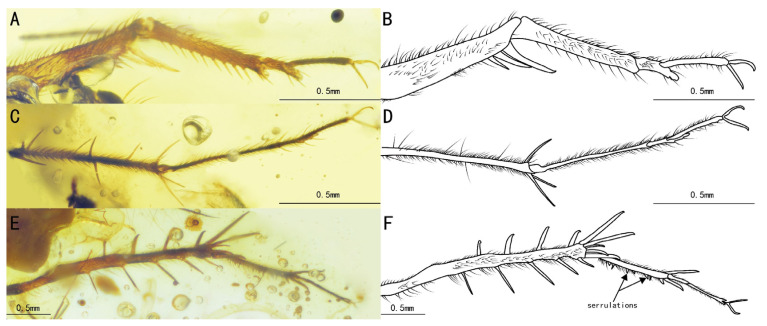
Details of *Tricalcaratus longilineus* sp. nov.: (**A**) photograph of the prothoracic leg, (**B**) line drawing of the prothoracic leg, (**C**) photograph of the mesothoracic leg, (**D**) line drawing of the mesothoracic leg, (**E**) photograph of the metathoracic leg, and (**F**) line drawing of the metathoracic leg.

**Figure 7 insects-15-00442-f007:**
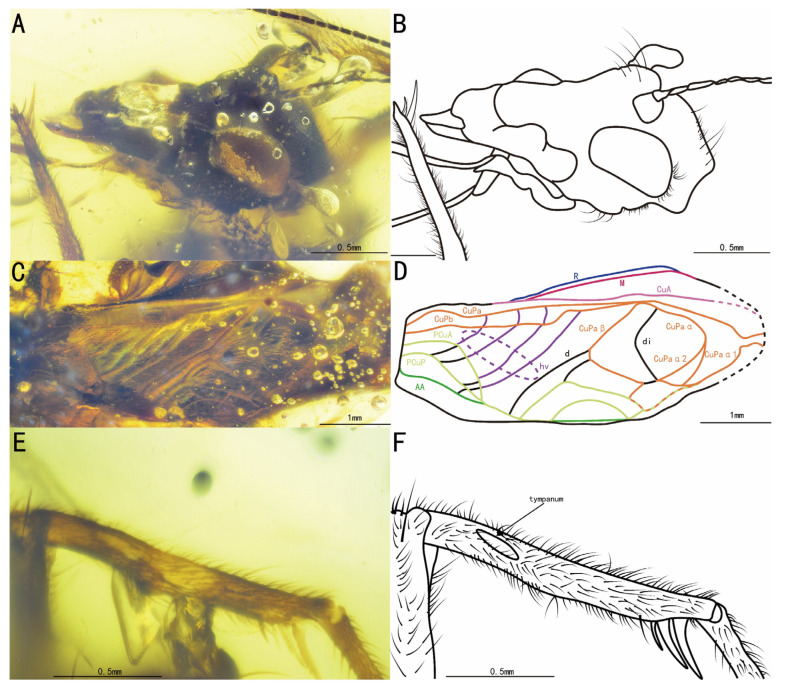
Details of *Tricalcaratus longilineus* sp.nov.: (**A**) photograph of the head, (**B**) line drawing of the head, (**C**) photograph of the tegmina, (**D**) line drawing of the tegmina, (**E**) photograph of the prothoracic leg to show the tympanum, and (**F**) line drawing of the prothoracic leg to show the tympanum.

**Figure 8 insects-15-00442-f008:**
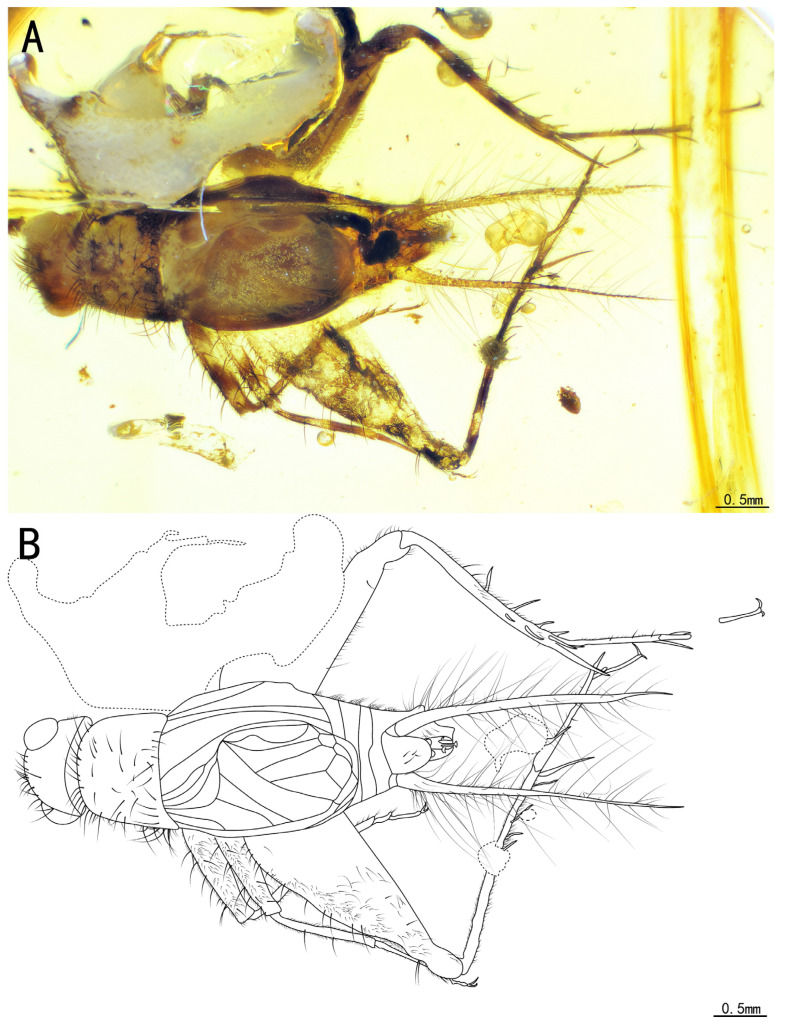
Habitus of *Birmaninemobius hirsutus* Xu et al., 2020, SICAU(A)-163: (**A**) photograph of dorsal view and (**B**) line drawing of dorsal view. Dotted line indicates obscured parts.

**Figure 9 insects-15-00442-f009:**
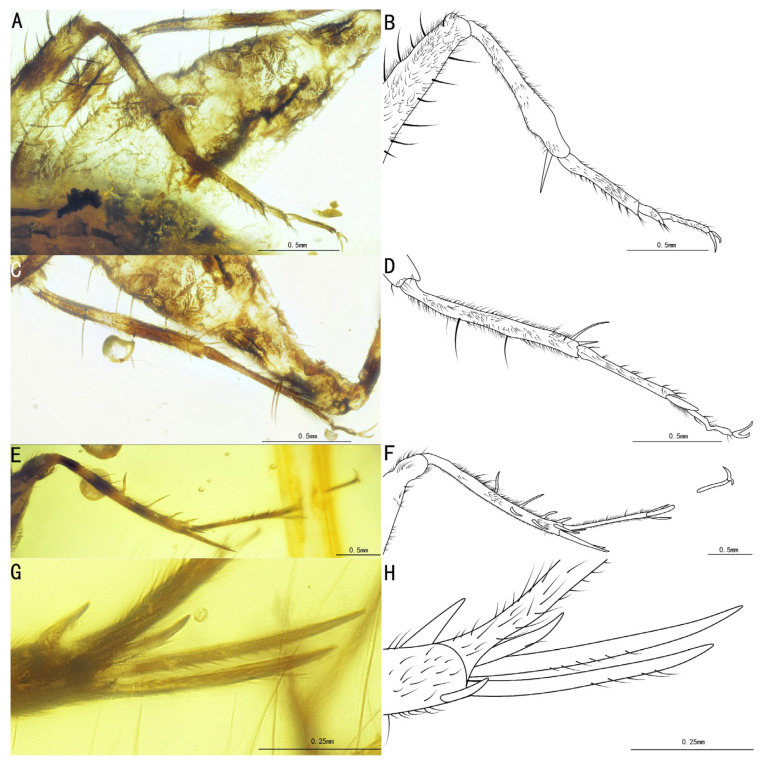
Details of *Birmaninemobius hirsutus* Xu et al., 2020, SICAU(A)-163: (**A**) photograph of the prothoracic leg, (**B**) line drawing of the prothoracic leg, (**C**) photograph of the mesothoracic leg, (**D**) line drawing of the mesothoracic leg, (**E**) photograph of the metathoracic leg, (**F**) line drawing of the metathoracic leg, (**G**) photograph of the apical spurs on the metatibia, and (**H**) line drawing of the apical spurs on the metatibia.

**Figure 10 insects-15-00442-f010:**
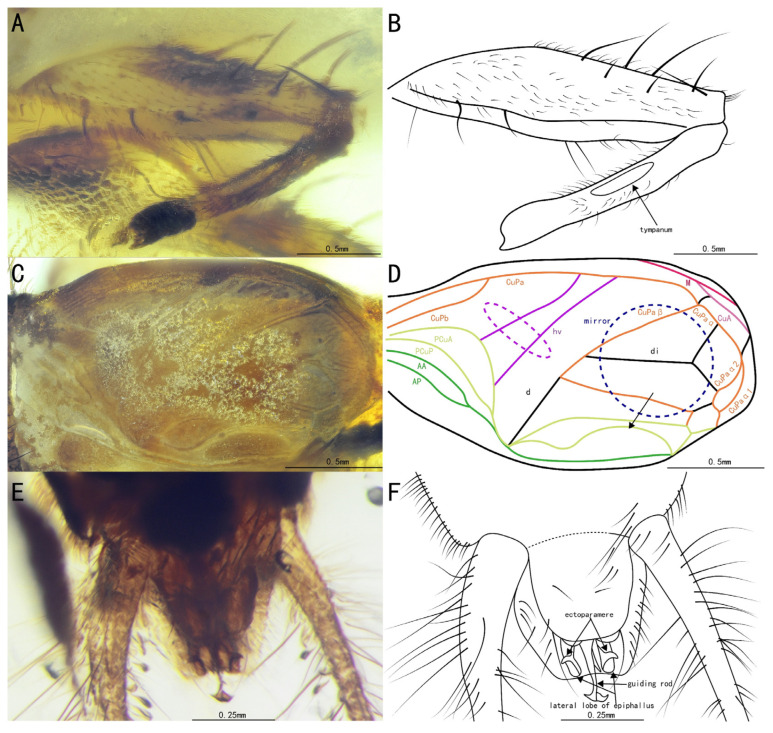
Details of *Birmaninemobius hirsutus* Xu et al., 2020, SICAU(A)-163: (**A**) photograph of the prothoracic leg to show the tympanum, (**B**) line drawing of the prothoracic leg to show the tympanum, (**C**) photograph of the tegmina, (**D**) line drawing of the tegmina, (**E**) photograph of the terminal abdomen, and (**F**) line drawing of the terminal abdomen.

**Figure 11 insects-15-00442-f011:**
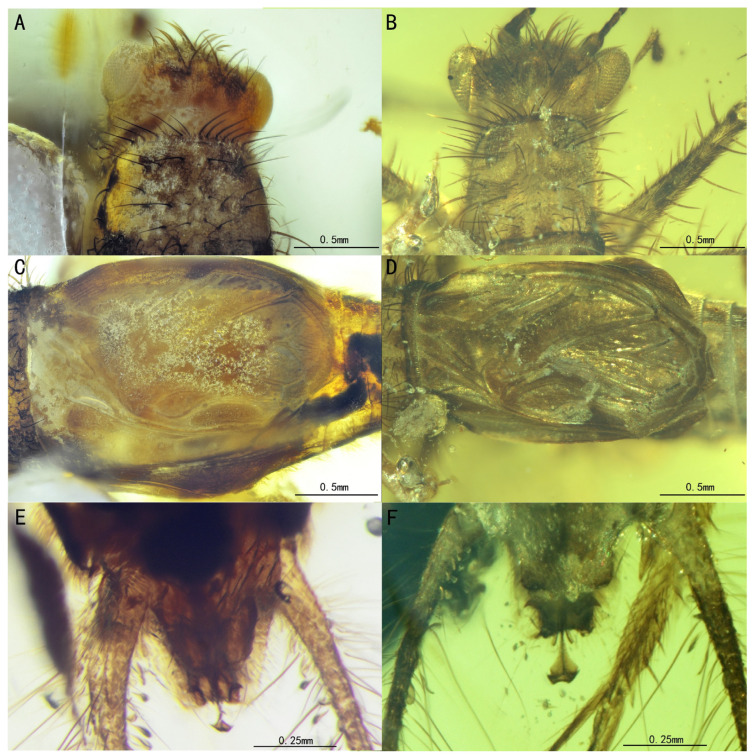
A detailed comparison between SICAU(A)-163 and NIGP172331: (**A**) head of SICAU(A)-163, (**B**) head of NIGP172331, (**C**) tegmina of SICAU(A)-163, (**D**) tegmina of NIGP172331, (**E**) terminal abdomen of SICAU(A)-163, and (**F**) terminal abdomen of NIGP172331.

## Data Availability

No new data were created or analyzed in this study. Data sharing is not applicable to this article. Nomenclatural Acts: urn:lsid:zoobank.org:act:053F9F92-E41B-4473-BB48-7E08DA0EFF03; urn:lsid:zoobank.org:act:7B525084-784D-474A-A364-57EF068D391D; urn:lsid:zoobank.org:act:0E738C42-EC95-4480-8208-2754F1491CF8; urn:lsid:zoobank.org:act:D5CE79E8-B3E7-41B5-8BAC-F93D189A5B60; urn:lsid:zoobank.org:act:02B966D9-3431-47F8-993B-075198209B58.

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
