# Peer review of "New Genera and Species of Trigonidiidae (Orthoptera: Grylloidea) from the Mid-Cretaceous of Myanmar with a Redescription of Birmaninemobius hirsutus"

_insects, 2024, doi:10.3390/insects15060442_

Round 1
Reviewer 1 Report
Comments and Suggestions for Authors
Dear authors,
Your paper entitled " New Genera and Species of Trigonidiidae (Orthoptera: Grylloidea) from Mid-Cretaceous of Myanmar with a Redescription of Birmaninemobius hirsutus" is an important contribution to the scientific community, mainly for Grylloidea systematics and fossil records.
However, there are some minor issues that I recommend to be reviewed.
1-In the first line of the introduction, you mention that Trigonidiidae is sister to all other Grylloidea. That is not true. Mogoplistidae is sister to all Grylloidea, including Trigonidiidae. See Chintaun-Marquier et al 2016 (Cladistics) and Ferreira et al. 2024 (Syst. Ent.) .
2- In the third line of the intro, the most important reference for Orthoptera numbers, classification, and distribution is the Orthoptera Species File (Cigliano et al. 2024).
3- In the last paragraph of your material and methods (page 2, line, 65) I suggest you cite Josse et al 2023 (Zoosystema). This is the most recent and complete paper on cricket wing venation. I believe you are following their terminology already.
4- Diagnosis of Palaeotrigonidium concavoculus (page 3, line 28): Besides the eye's morphology, all the other characters are present in several other Trigonidiidae crickets. It would be important to include more exclusive characters or set of characters in this diagnosis.
5- This is a recommendation for all your descriptions: Remove the article “the” and other particles from the descriptions. They are indicated through your descriptions in the file attached. This will make your description more synthetic and less repetitive. Moreover, this is a recommendation for taxonomic descriptions. I suggest you compare it with other taxonomic descriptions.
There are some other comments and corrections in the attached file.
Best regards.

Dear Authors and Editor,
I am not a English native speaker. However, some corrections must be made in the manuscript. Most of them are indicated in the attached file. I would like to emphasize again the use of too many articles in the descriptions. This is not recommended in taxonomical descriptions. I suggest the authors review this issue, and maybe compare it with other taxonomic papers.
Reviewer 2 Report
Comments and Suggestions for Authors
Dear authors,
Your manuscript is quite good in terms of subject matter, which is obviously of interest.
However, some improvement suggestions are needed to increase the value of your work.
Lines 45-47: It is not clear why you redescribed the species Birmaninemobius hirsutus. What is the reason and importance of this study. Be more explicit and detailed in comparison with what already exists, why you focused on this species as long as you mention that it is already known. In fact, you must reinforce the value of the study through justification.
Question: Can it be one of the explanations that in the specialized literature the systematic framing is ambiguous? Some say it belongs to the Nemobiinae, but others dispute it with evidence that it could rather belong to the Trigonidiinae stem group?
Taken as a whole, the Introduction is little supported by the references of other studies. You should elaborate more and attribute more sources as there are, especially for Birmaninemobius hirsutus.
Line 50: It would be good to provide a brief history of these specimens stored at the University. More precisely, the estimated year of storage and under what circumstances did they arrive at the university? These amber specimens are part of a collection somehow and to whom does the collection belong? Something is not very clear in the circuit of insect samples.
As a whole/ Materials and Methods: It would also be advisable to present the degree of damage of the analyzed specimens because in the results you bring to attention photos with evidence of them with various degrees of damage it seems. Either here or at Results in the legend of the figure. In the present form, there is no unitary line in the evaluation of the specimens, the missing parts being practically unexplained. Then how did you clearly determine each specimen?
On results/all figures: Mention somewhere in the legend what the dotted line means, you probably refer to the damaged part of the sample analyzed; but even so it is good to say this so that all readers understand
To references/as a whole: Please keep the same single line for all references, for example in Ref 1 write the authors like this: Chintauan-Marquier, I.C., Legendre, F., Hugel, S., Robillard, T., Grandcolas, P.,... on when in ref 2 you write like this: Capinera, J.L.; Scott, R.D.; Walker, T.J and at Ref 15 write like this: Liu, Y.J., Yu, Z.Y. & He, Z.Q. A. Read the instructions and write according to the requirements, either with ", " or "; " or "&" between authors
Line: 460: If you add sources (as I recommended) to the Introduction, please detail them in the Reference List
Kind regards,
R
Reviewer 3 Report
Comments and Suggestions for Authors
With a redescription of Birmanimobius hirsutus (hirsutus? Is this a kind of Lotus? A plant or insect?), there are too many things in the New genera and species of Trigonidiidae (Orthoptera: Grylloidea) from the Mid-cretaceous of Myanman. For a publication that focuses on new findings, either on genera, species, or (re)-descriptions, eleven figures plan is too many. Here, an attempt should be made to select a story and concentrate on one that might be interesting to Insects. If the narrative were written better, it might fit better in Arthropod Structure and Development.
Another crucial aspect is the writing style. For example, the simple summary is anything but simple. The purpose of writing such an abstract is to make Trigonidiidae (Orthoptera: Grylloidea), Trigonidiinae, and Nemobiinae understandable to a broad audience, but I don't think they are this way. People are even unaware of the importance to science of how we study extinct insects. People may not be aware of the existence of Birmaninemobius hirsutus, the definition of a holotype, or the significance of the reexamination. A Trigonidiidae: What is it?
I'm not sure if the scientific abstract is written more effectively. It perhaps mentioned three “newly” discovered species from Myanmar that were folded into amber. Myanmar is where? Probably the first figure should be a map of the exploration site. Why is it that these species are found there so important? Can a new species be established with just three specimens? Since the study only discusses morphology, it is unclear from the abstract what exactly the remarkable morphological traits would allow for the emergence of a new species. Wouldn’t it be more appropriate to simply say that we found a gryliid specimen that lacks subfamily assignments? Which ones? A sentence such as "A new specimen and recheck of the holotype lead to a redescription of Birmaninemobius hirsutus, another known species" (lines 21–22) is not very appropriate for two reasons: 1) Birmaninemobius hirsutus is not well known (is it related to Qiongqi crinalis? ), and 2) one specimen is insufficient for a recheck.
I found the text in use too redundant with text from internet and Science News in the field of amber insects. https://www.sci.news/paleontology/qiongqi-crinalis-12295.html#google_vignette
Naturally, the authors are free to use the provided material, but they ought to make an effort to make their writing original and avoid duplicating what has already been published in Wei Yuan et al. 2023. A new genus and a new species of Trigonidiidae (Orthoptera: Grylloidea) from north Myanmar amber. Zootaxa 5330 (1): 141-146; doi: 10.11646/zootaxa.5330.1.9 In addition, especially if the study is in the continuity of Yuan et al. (2023), they should follow more carefully the information which analyzes much more organs and tissues, morphology, shape and structure of this arthropod.
How do the results differ from the first ground cricket (Orthoptera: Trigonidiidae: Nemobiinae) from mid-Cretaceous Burmese amber (Xu et al., 2022)?
I don't know a lot about the supplies and techniques used in fossil insect analysis. Still, I'm sure there's a lot more to offer than just picture information (see lines 56–62). Is it really that easy to analyze fossil insects? It is the authors' responsibility, not the list of references', to characterize the fossils. How was the specimen's age determined? Using data from radioisotope, biostratigraphy, and/or zircons? Not only do we not know where the specimen was collected, but we also don't know how, when, or close to what it was. Even the method of preserving and storing the specimens is not described and/or provided.
For publication, a detailed electron microscopy analysis ought to be carried out at the absolute least.
Material and methods is twenty lines with two paragraphs of terminology.
The results (refer to lines 71–80) indicate that the article's target audience is not Insects-MDPI. Diagnosis, Etymology, Description, Redescription, are not outcomes. Drawings and photos don't yield enough information either, especially since none of them accurately represent points of legends. Even the location of the crucial point on the figure is not shown to us. A few of the images (2A, 3A, 4A, 4G, etc. and even more) are of fairly low quality. There must be technology that can be used to get the pieces of amber out of the insect image.
The various features of heads, legs, abdomens, and wings should be measured, and this requires the addition of some tables. Why aren't the oviposition gland or entire antennae mentioned? The antennae are repeatedly characterized as filiform. If their sensilla were like legs, there might be more to say about them.
Here, no conclusion(s) should be drawn. If you want to draw a conclusion like "These new findings, new species, genera, and re-evaluation," please compare the data with what is already known about amber gryllids and offer a more thorough, rigorous, and definitive analysis. This text and the pictures lack context and provide no comparative data, so it is very unclear.
Comments on the Quality of English LanguageHere, an attempt should be made to select a story and concentrate on one that might be interesting to Insects. If the narrative were written better, it might fit better in Arthropod Structure and Development.
The simple summary is anything but simple.
I'm not sure if the scientific abstract is written more effectively.
Round 2
Reviewer 3 Report
Comments and Suggestions for Authors
The authors failed to respond to the comments or even attempt to comprehend the purpose of a summary for a broader audience—that is, the general public—who is ignorant of taxonomy and the necessity of examining insects preserved in amber. I continue to think that the number of figures is too high to publish in the journal and that additional analysis, such as electron microscopy, should be done. In the event that these data (EM, molecular data, morphology, etc.) are supplied by some other studies, the text ought to be composed with greater caution.
Author Response
This is a taxonomic work and prepare for the section"Insect Systematics, Phylogeny and Evolution" of this journal, we have to write it follow the insect taxonomic and systematic rules and style. we can not write a story as his/her suggestion. we try to show the most detailed structures of species for readers, that's why we provide 11 figures in the paper. we try to scan the ambers, but failed, it is very hard to get a result of scanning from orhoptera ambers of Myanmar.